# Refinements in Clinical and Behavioral Management for Macaques on Infectious Disease Protocols

**DOI:** 10.3390/vetsci11100460

**Published:** 2024-10-01

**Authors:** Lauren Drew Martin, Jaclyn Shelton, Lisa A. Houser, Rhonda MacAllister, Kristine Coleman

**Affiliations:** Division of Comparative Medicine, Oregon National Primate Research Center, Beaverton, OR 97006, USA; sheltoja@ohsu.edu (J.S.); houserl@ohsu.edu (L.A.H.); macallis@ohsu.edu (R.M.); colemank@ohsu.edu (K.C.)

**Keywords:** social housing, positive reinforcement training, SIV, procedure cage

## Abstract

**Simple Summary:**

Nonhuman primates (NHPs) on infectious disease studies can be challenging to manage both clinically and behaviorally. These animals may develop significant clinical signs as well as stress-related problems. Not only does this negatively affect the welfare of the animals, but it can also impact the validity of scientific outcomes. Thus, while collaboration between the veterinary and behavioral teams is critical to maximize animal welfare and research outcomes in any study, it is especially true in infectious disease work. In this review, we discuss some refinements to promote the psychological well-being and resiliency and to enhance the overall health of NHPs on infectious disease studies, which can help improve welfare and, in turn, scientific outcomes.

**Abstract:**

Providing optimal clinical and behavioral care is a key component of promoting animal welfare for macaques and other nonhuman primates (NHPs) in research. This overlap between critical areas of management is particularly important for NHPs on infectious disease protocols, which often have unique challenges. For example, traditionally these NHPs were often housed alone, which can have behavioral and clinical consequences. However, in the past decade or so, considerable effort has been directed at modifying procedures in an effort to improve animal welfare for this group of NHPs. In this review, we examine some refinements that can positively impact the clinical and behavioral management of macaques on infectious disease studies, including increased social housing and the use of positive reinforcement techniques to train animals to cooperate with procedures such as daily injections or awake blood draws. We also discuss ways to facilitate the implementation of these refinements, as well as to identify logistical considerations for their implementation. Finally, we look to the future and consider what more we can do to improve the welfare of these animals.

## 1. Introduction

Nonhuman primates (NHPs) have made significant contributions as a key animal model in infectious disease (ID) research breakthroughs throughout history [1,2,3]. NHPs, particularly macaques, continue to be critical models of HIV, Zika, Mpox, COVID, and *Mycobacterium tuberculosis*, in addition to a plethora of endemic and emerging infectious diseases [4,5,6,7]. ID studies can be more clinically and behaviorally challenging depending on the agent of interest, route of infection, types and frequency of samples collected, length of study, and study endpoints. Animals in ID studies are often more demanding to manage than others, as infection with the agent of interest, as well as certain antibody treatments, may result in significant clinical signs including, but not limited to, decreased appetite, diarrhea, nausea, decreased activity, fever, lethargy, skin irritation, hepatotoxicity, renal toxicity, and localized or generalized inflammation (personal observation). Further, these animals may be at a higher risk than others for developing stress-related behavioral problems, particularly if they must be singly housed for all or part of their study, as reviewed in [8]. Thus, while collaboration between the veterinary and behavioral teams, along with the research group, is critical to maximize animal welfare and research outcomes in any study, it is especially true in ID work.

While the physical and behavioral complications associated with ID research can compromise welfare, they can also negatively impact the validity of scientific outcomes. It has long been established that highly stressed animals are not reliable subjects for most scientific studies except, perhaps, for those examining the effects of stress. Psychosocial stress and compromised welfare can alter the hypothalamic–pituitary–adrenal axis, as well as immunological function in NHPs [9,10]. Further, because individuals vary widely in their physiological and behavioral responses to stress [11,12,13], they can increase overall experimental variability [14,15]. As one of the Three Rs [16] that guide research approaches, refinements that promote the psychological well-being and resiliency of the NHPs, thus enhancing overall health throughout an ID study, can help improve welfare and, in turn, scientific outcomes. Decreasing inter-individual variability not only increases the validity of the research but may result in a reduction in the number of subjects required to reach statistical significance in experimental protocols, which is another of the Three Rs.

In this report we will explore current social housing, positive reinforcement training, and behavioral assessment approaches to refine the care of NHPs on ID studies through an integration of veterinary and behavioral strategies and interventions. We will also discuss considerations for implementing these strategies, including key stakeholders to engage in the process. The goal of this report is to provide a thorough, although not exhaustive, overview of current practices and future directions for veterinary and behavioral collaboration to enhance the care of NHPs on ID studies.

## 2. Social Housing

### 2.1. Pair Housing Macaques with SIV

Social housing, including pair housing, is widely accepted as the best way to promote health and well-being for research macaques (*Macaca* sp.) [17,18]. Pair-housed macaques exhibit more species-typical behaviors [17], fewer abnormal behaviors [19,20,21], fewer signs of stress [22,23,24,25], and different immunological responses [25] compared to their singly housed counterparts. Further, social housing can improve the NHPs’ resiliency to aversive stimuli and other stress. For example, paired macaques show fewer abnormal behaviors [26] or behavioral indicators of anxiety [27] when faced with stressful stimuli compared to those housed individually. Because many physiological processes are significantly affected by stress, social housing—or the lack thereof—may impact research variables. 

While social housing has for many years been the default for NHPs on most scientific projects, it is somewhat less commonplace in ID protocols, particularly for studies in which viral transmission across subjects is a concern, such as simian immunodeficiency virus (SIV) acquisition. The primary reason provided for social housing exceptions for these kinds of projects includes the risk of viral spread through saliva or blood, as may happen during agonistic encounters. For example, monkeys involved in HIV cure studies that include the suppression of SIV replication with antiretroviral therapy (ART) treatment are often pair-housed for part of the study but separated prior to and during SIV infection to prevent viral transmission between partners. At many, but not all, facilities, these monkeys may be reunited once the animals are no longer viremic. 

As discussed above, singly housing NHPs can increase stress, which can lead to increases in stress-related illnesses and behavioral issues. Several studies have found that singly housed rhesus macaques (*Macaca mulatta*) are more likely to develop abnormal behaviors, including self-injurious behavior, than socially housed macaques [19,20,21]. There have been fewer studies examining the effects of single housing on research outcomes in infectious disease protocols. However, recent studies have demonstrated that the stress of single housing may impact the pathogenesis of SIV infection in pigtailed macaques (*Macaca nemestrina)* [28,29]. In a retrospective study, the authors compared parameters of SIV infection (e.g., viral load, CD4 T-cell decline) between singly and socially housed macaques. All subjects were inoculated intravenously with the same stock of SIV inoculum and began antiretroviral therapy (ART) 12 days after infection. Singly housed monkeys were housed without a partner for approximately 2 months prior to inoculation and throughout the study, while socially housed NHPs were paired with 1–2 conspecifics over that same time period. Singly housed macaques had higher viral loads, greater CD4 T-cell declines, and greater CD4 and CD8 T-cell activation during acute SIV infection compared to socially housed conspecifics [28]. Monkeys housed alone also showed a reduced expansion of monocytes and a suppression of platelet activation after inoculation [29]. Importantly, the results displayed by the socially housed macaques more closely aligned with what is seen in humans than those in the singly housed animals. For example, as expected, all macaques experienced a decline in CD4 T-cells during the acute infection phase. However, the magnitude of the decline in socially housed animals was approximately 2-fold, which is analogous to humans during primary HIV infection [28]. The singly housed animals, in contrast, had a 3-fold change. These results suggest that social housing improves the translational value and reproducibility of the data [28]. 

Even relatively short periods of socialization can help mitigate the negative impacts of single housing on ID protocols. In a recent study [30], investigators examined the effect of changes in housing on cell activation and vaccine-mediated immune responses in juvenile rhesus macaques. Monkeys were moved from large social groups to caged housing in one of three conditions for the 10–14 weeks of the study. One group of macaques was paired after removal, one group was singly housed, and the last group was paired for 5 weeks and then housed alone. Monkeys received a measles vaccination after being moved to caged housing. The authors found greater CD8 T-cell expansion and a higher expression of activating B-cells in the animals that were singly housed the entire time than those that were pair-housed before being singly housed, suggesting that even brief pair housing can provide a buffer to the stress experienced by singly housed animals [30]. 

Not only does social housing seem to affect translational value and reproducibility, but it also helps reduce the variability in experimental outcomes. Guerrero-Martin and colleagues [28] compared the standard deviations of the data generated from singly and socially housed animals and found that data from singly housed animals were significantly more variable. As detailed above, reducing data variability can reduce the number of animals needed, which is also an important tenet of the use of animal models in research [15].

Despite the benefits of social housing for the animals and research outcomes, it is not without risk. Cage-mates can cause injury if they are not compatible, and even without overt aggression, incompatibility can lead to stress for one or both partners. Thus, it is important for everyone on the team, including the behavioral and clinical staff, to carefully pick partners, and to closely monitor pairs for signs of incompatibility [31]. It can be somewhat more time-consuming to provide clinical care to paired animals, particularly if only one member of the pair needs treatment, as the clinical staff need to identify the animals and make sure that the right animal gets the therapy. In some cases, cage-mates may need to be temporarily separated from each other for treatment, which can take additional time. However, there are ways to mitigate this potential challenge, such as dye marking each animal or using positive reinforcement techniques to train animals to come to the front of their cage for treatment (see below). 

### 2.2. Socialization of Infants

The past decade has seen an increase in the number of SIV studies utilizing infant macaques to model HIV vertical transmission and cure research. Such studies often rely on nursery-rearing subjects and require a daily administration of ART. Although commonly used for juvenile and adult monkeys, continuous pairing of two infants in the nursery is not generally considered the best way to house young monkeys. This kind of housing can lead to excessive clinging between the infants [32], which makes it challenging to separate the animals from one another when needed. At the Oregon National Primate Research Center (ONPRC), clinical, behavioral, and husbandry staff worked closely together to design specialized infant cages that allow infant rhesus macaques to be housed in small groups (e.g., 4–6 individuals) while still allowing temporary separations for feeding or research-related procedures (Figure 1). We see less clinging behavior when infants are housed in small groups than pairs or triplets (Houser, unpublished data). In addition, separating infants during feeding appears to acclimate them to being alone for short amounts of time and thus makes separations for other reasons easier (personal observation). Cages provide vertical and horizontal access for the infants and allow swings and manipulatable objects. Importantly, transmission of SIV has not been observed in infants housed with 4–5 similarly aged conspecifics (N Haigwood, personal communication), suggesting that group housing of young macaques on ID protocols is possible. Housing the infants in small groups also reduces the potential for single housing if an animal needs to be removed from the study.

### 2.3. Instrumented Animals

Despite the benefits of social housing, NHPs with chronic implants, such as cranial implants or chronic intravenous catheters, have historically been individually housed, due to concerns about conspecific trauma associated with pair housing and damage to the implanted materials. This trend is changing, as researchers show that pair housing instrumented macaques is possible. Several studies have provided evidence that monkeys with devices, such as cranial implants [17,18,33] and vascular access ports [34,35], can be pair-housed without adverse consequences. Still, some instrumented animals are still widely singly housed, including those with chronic indwelling catheters, as may happen in ID studies when frequent infusions and/or blood samples are needed (e.g., for studies involving stem cell transplantation or chimeric antigen receptor (CAR) T-cell therapy administration). 

One area of HIV cure research is aimed at establishing a macaque model of stem cell transplantation to functionally cure HIV e.g., [36], as was the case with Timothy Brown (the “Berlin Patient”), the first person in the world to be cured of HIV following stem cell transplantation [37]. Because stem cell transplantation can have potentially significant complications, it requires additional clinical monitoring and care, including chronic catheterization [36]. Such catheterization often results in single housing for the animals. At the ONPRC, a team of veterinary, behavioral, and research personnel work together to maintain monkeys on these studies. This team noted that chronic catheterization combined with separation from partners appeared to be associated with inappetence that required clinical intervention, and so the feasibility of keeping the animals paired while one partner was catheterized was examined [38]. The subjects were two female Mauritian cynomolgus macaques (*Macaca fasciularis*) housed in isosexual pairs. Both NHPs had been cohoused with their partner for at least 50 days prior to catheterization. Trained behavioral staff monitored the pairs and determined that they were highly compatible.

Prior to catheterization, the monkeys were acclimated to the jacket and catheter protection system (CPS). The study animal was fitted with the mesh jacket and staff monitored the NHP, both in-person and remotely, to determine whether the partner manipulated the jacket. Animals were also provided with additional enrichment to redirect them from manipulating the jacket. Once staff were comfortable that the animals were not manipulating and damaging the jacket, a tether line was added. This step helps acclimate the subject to the added rigidity of movement around the cage. The acclimation period provides time to allow the novelty to wear off, as both animals may explore and examine the equipment. Maintaining full pair status through the acclimation process conditioned both animals to the jacket and CPS before the chronic catheter was placed. 

The pairs were temporarily (approx. 1 h) separated with a mesh or grooming contact slide for both feedings and enrichment so that food intake could be monitored. The animals continued to display prosocial behavior (e.g., huddling, grooming, lipsmacking) after surgery (Figure 2). Non-catheterized animals investigated their partner’s catheter protection system, but did not damage the jacket, chronic catheter, or CPS. Importantly, the catheterized monkeys had a better appetite than previous NHPs who had been separated from their partners, suggesting that pairing helped reduce stress and improved welfare for the animals [38]. 

Since that time, three additional animals (1F, 2M) remained pair-housed while undergoing the stem cell transplantation process. While pair housing NHPs with chronic catheters may not be appropriate in all situations (e.g., new pairs), it is possible for some animals with chronic catheters. Using an established pair that previously displayed affiliative behaviors (and lack of aggression) reduced the risk associated with pairing while catheterized. 

## 3. Positive Reinforcement Training

ID animals often undergo various procedures while on study, including injections (e.g., subcutaneous ART injections, sedation events), blood sampling, and measuring oxygen saturation levels. They may also need to present specific body parts for veterinary exams. As they may be more likely to become ill, they may require veterinary exams relatively frequently, which can also be stressful. One way to help reduce the stress surrounding these procedures is training animals to cooperate with procedures using positive reinforcement training (PRT). PRT techniques are a form of operant conditioning e.g., [39], in which subjects are rewarded with something desirable (e.g., a food treat) for performing specific behaviors on command. In PRT, the subject is presented with a stimulus (e.g., a verbal cue), responds by performing a specific behavior (e.g., move to the front of the cage and remain stationary), and is provided with reinforcement (e.g., a food treat) when the specific behavior has been completed (see Pryor [40] and Laule, Bloomsmith [41] for reviews). The use of PRT is recognized as an important tool for promoting well-being in captive species, including NHPs [42,43,44,45,46]. Macaques and other NHPs have been successfully trained to perform various husbandry or clinical tasks, including moving to a new part of an enclosure (i.e., “shifting”) [47], presenting a body part for injection or another procedure [42,48], taking oral medications [35,49], and remaining stationary for blood sampling [46,50].

There are many welfare benefits to using PRT, which aid both clinical and behavioral management. By allowing individuals to cooperate with various procedures, PRT can reduce the stress associated with these procedures [41,42,51], which can reduce the chances of stress-induced illnesses (e.g., prolapses, diarrhea) and behavioral problems. It can also increase well-being by decreasing boredom and increasing mental stimulation for subjects [41,52]. Trained animals are often more cooperative and thus easier—and safer—to work with than are untrained animals [53]. While it may take time to train NHPs for specific tasks, the time invested will likely result in significant time savings when conducting procedures with trained subjects [45,47], which can help clinical as well as research and husbandry staff.

Like social housing, PRT has been shown to decrease the stress associated with experimental procedures as well as minimize potential confounds, thus reducing the experimental variability in macaques [15]. For example, Graham and colleagues [35] found that NHPs trained to cooperate with tasks such as presenting a limb for access to an indwelling vascular access port (VAP) showed a significant reduction in stress compared to when animals were either chemically (e.g., sedation) or physically (e.g., primate chair) restrained. Further, they found that while there was a time investment with the initial training, once the animals were trained, it took significantly less time to perform the procedures [35]. While there are few published reports examining PRT for animals on ID protocols, a recent study has shown that PRT can mitigate stress associated with SIV infection in pigtailed and rhesus macaques, including lower cortisol, reduced viral loads, decreased T-cell activation, and increased innate immune response [54].

PRT can also directly facilitate the clinical care of animals on ID studies. Monkeys can be trained for behaviors such as remaining stationary for injection, separating from their partner for medication administration, and presenting body parts for clinical examination. For example, macaques on ID studies may develop conditions including psoriasiform dermatitis or allergic contact dermatitis with a psoriasiform histologic pattern, which can result in xerosis, hyperkeratosis, fissures, and ulcers on the palmar and plantar surfaces of hands and feet, ischial pads, or scrotum (personal observation). The standard treatment for this condition is steroids; however, such steroid treatments are contraindicated on many ID studies. At the ONPRC, we train animals with this condition to present hands or other body parts to allow cage-side administration of topical medications (Figure 3). This training, which typically takes 2–3 weeks to accomplish, has helped animals with dermatitis remain on their ID protocols. 

Ideally, all animals would be trained to voluntarily cooperate with procedures such as remaining stationary for ART injections and other procedures common in ID studies. For example, NHPs can be trained to come to the front of the cage and present a limb for VAP manipulation [35] or blood draw [46]. However, this may not be feasible at every institution or with every animal. Recent equipment refinements have facilitated these procedures. For example, ONPRC clinical and behavioral staff modified a “procedure cage”, a removable cage that attaches to the exterior of an NHP’s home enclosure, for use with daily ART injections. The NHPs can be trained to enter the procedure cage and remain stationary (Figure 4). Once in the procedure cage, the animals are confined, rather than restrained, although they can be restrained through a reversible squeeze mechanism if necessary. The procedure cage confines the animals by temporarily limiting their access to parts of their cage, similar to how dog crates confine, but do not restrain, canines. While originally designed for ART injection, we included access ports with removable coverings to allow more flexibility in the kinds of procedures that can be performed in the procedure cage. Monkeys can be trained to offer limbs for procedures including blood draws or measuring oxygen saturation levels (Figure 5). The ports also allow for devices to be inserted into the cage for procedures such as conscious ultrasound.

In collaboration with a local vendor (Carter2Systems, Inc., Hillsboro, OR, USA) the ONPRC has also developed a procedure cage for infants receiving daily ART injections. Very young infants are often held for procedures such as ART injections; however, handling older infants can eventually present safety risks for staff. The type of restraint typically used for adult macaques (e.g., a squeeze-back mechanism on cages) often does not work well for very small animals. The infant-specific procedure cage has an insert that is fitted with a fleece-covered pad resembling their stuffed surrogates to provide comfort. The infants are trained to enter the procedure cage and hold onto the pad in the prone position while the curved sides fold gently around them to expose their backs for their injections. Infants adapt to this relatively well, and the staff providing ART injections have reported that the animals are easier to work with when in the procedure cage. 

While PRT might not be feasible at all institutions, acclimating animals to equipment, procedures, and people takes less time and is beneficial to the animals. Unstructured human interactions, such as providing treats or talking to the animals, have been shown to reduce abnormal behavior, increase species appropriate behaviors, and improve well-being for a variety of primates, including marmosets, macaques and chimpanzees [55,56,57,58]. Importantly, these relationships can also promote coping skills [59] and help mitigate stress reactivity towards novel objects or situations. The amount of time spent in positive interactions does not have to be great to be effective; simply handing out treats for a few minutes several times a week reduced indicators of stress in cynomolgus macaques and being in a room while occasionally handing out treats reduced abnormal behavior and reactivity in rhesus macaques [60]. These kinds of positive human–animal relationships can help reduce stress and improve well-being for the monkeys, something that is important for animals on ID protocols. Further, these interactions can also improve the morale and decrease the burnout of care staff working with monkeys on ID protocols, which, in turn, can also promote welfare for the animals [60]. 

## 4. Animal Assessments

Another refinement to ID practices is ensuring that the right animals are assigned to the right project, thus setting up each individual for success. ID projects may involve frequent sampling or injections, and not all NHPs handle these stressors the same way. Factors such as age, sex, and temperament can affect stress sensitivity of NHPs [61,62]. These differences can affect their susceptibility to illness and the development of stress-related abnormal behavior [19,20]. It can also influence how quickly they can be trained for injections and other procedures. For example, animals that are highly inhibited may not perform training tasks as quickly as other animals [63]. At the ONPRC, we evaluate the temperament, stress sensitivity, motivation, and trainability of a majority of NHPs prior to assignment. The assessments provide information about an animal’s suitability for a specific project. Individuals that are highly stress sensitive or fearful may not handle daily ART injections well. It is not always possible, or even desirable, to pick only the most trainable subjects for ID or any studies. However, these assessments can also identify animals that may benefit from additional acclimation and training time.

## 5. Discussion 

While the refinements in clinical and behavioral management discussed above are designed to improve both animal welfare and research outcomes, there are several factors that should be considered prior to their use. Each facility and research program has unique operational and organizational logistics that impact the implementation of these strategies and interventions. To garner support from invested parties, key stakeholders should be included in discussions of refinements at an early stage to ensure effort is not wasted, to raise and address potential questions around logistical challenges, and to help with brainstorming ways to overcome challenges. Key groups to consider, in addition to the veterinary and behavioral teams, include the principal investigator and research staff, environmental health and safety staff, the IACUC or commensurate oversight body, animal care staff, the colony management or animal allocation team, and program leadership. Even inclusion of the facilities maintenance staff can often be insightful.

Interfacing with the research staff during the study development and grant-writing phase provides opportunities to discuss various options for the implementation of the refinements addressed here. Investigators may be able to adjust their study designs to accommodate behavioral management strategies, such as social housing or the incorporation of PRT, and they may also be able to include any costs necessary to implement these options into the initial grant. Discussions should continue throughout the course of the study, allowing for the review and revision of the strategies and their impacts on animal care and research aims. Developing the researchers as partners in the implementation of these strategies and interventions encourages buy-in and continued exploration of these options with future studies. Pilot studies may be requested to show how strategies impact, and possibly even improve, data; these studies may provide objective measures and can strengthen a researcher’s and institution’s willingness to investigate refinements. This process of the communication and inclusion of refinement strategies should be developed and integrated as part of the culture at a facility, guiding interactions between animal care and research staff and promoting continual improvements in both animal welfare and research outcomes.

Collaboration with environmental health and safety (EHS) through the assessment and implementation of refinements in infectious disease studies can mitigate potential negative impacts on staff. EHS involvement can also help to ensure institutional policies are followed or revised as needed to accommodate suggested refinements. The inclusion of these stakeholders provides an additional check that ensures human safety by determining if strategies (e.g., PRT and awake procedures) are increasing the potential danger of exposure to staff (e.g., blood collection from an awake animal vs. a sedated animal). If hazards are identified, EHS personnel can assist in brainstorming and developing processes to mitigate hazards while still achieving study goals (e.g., additional PPE and engineering controls).

While all refinements should be shared and discussed with the facility Attending Veterinarian (AV), any strategies directly related to research manipulations should be addressed in the IACUC protocol and reviewed by the IACUC or commensurate oversight body. Alternatively, it may be appropriate to address such refinements in the animal care program’s NHPs behavioral management plan; however, this is at the discretion of the AV. It is important that the IACUC understands the “why” behind the strategies and the impact they have on ongoing and proposed research. The IACUC can be an important advocate for pursuing and implementing approaches that improve animal welfare. 

Staff buy-in is critical to the success of many of these refinements, particularly if they involve increased staff effort to initiate and/or maintain. Including husbandry, sanitation, clinical, surgical, research, and facilities personnel provides an opportunity for education and again ensures they understand the “why” behind the refinements. Implementation may involve more work for them but will result in healthier and more behaviorally well-adjusted and resilient NHPs. These staff may also bring a different perspective to discussions and provide suggestions for improvements or cost savings that leaders may not have addressed. For example, facilities staff may have backgrounds in engineering and fabrication, so they can often suggest unique, “outside-of-the-box” ways to build and implement structural refinements.

Personnel responsible for colony management and animal allocation to research projects represent additional stakeholders to help implement and ensure the success of suggested refinements. This group can work closely with behavioral and veterinary staff to assess animals’ compatibility, trainability, and overall demeanor before allocation to research projects. They may advocate for research staff to assign paired animals to the same projects (allowing them to remain together) and to select resilient, well-adjusted NHPs for long-term or intensive projects. As described above, animals with self-injurious behaviors and those highly reactive to human interaction or interaction with other NHPs are typically not successful in ID studies requiring frequent manipulation and/or restraint. Partnering with the animal allocation team can assist in placing the most appropriate NHPs with the studies that set them up for successful clinical, behavioral, and research outcomes. 

Financial resources are often a constraint and must be considered as an aspect of the implementation of the reviewed strategies. As previously noted, working with the principal investigator during grant development may be an option to secure funding for both staffing and equipment. Including program leadership in the process of implementing refinement strategies helps to demonstrate how they enhance animal care and research outcomes as well as improve regulatory compliance and external group review (e.g., AAALAC). Facility leadership may also be a source of additional funding needed for refinement implementation. Leadership typically has a better understanding of the long-term goals and plans for the organization that may be impacted by or may directly impact the logistics of implementing refinements in NHP ID studies. 

As should be evident, it can take several weeks—or even longer—to appropriately plan for these types of studies. Structured planning meetings can make the process run more smoothly.

## 6. Conclusions

Garnering support from veterinary and behavioral staff around the refinement strategies discussed herein should be straightforward, given the beneficial animal welfare impacts. Enhanced psychological well-being improves animal health [64] by increasing the individual’s resiliency to disease and environmental change, as well as by reducing the severity of clinical outcomes from the administration of infectious agents to NHPs on ID protocols. With the expanding use of refinements, such as the ones discussed here, we are entering a new era of ID research in which the animal models may be even more translatable to the human condition.

## 7. Future Directions

While there have been many refinements over the last few years with respect to how NHPs on ID protocols are clinically and behaviorally managed, there is more we can do. It is critical for veterinarians and behaviorists to continue to work collaboratively to address the unique challenges of ID studies and to encourage the development of ID research staff as partners in this process of continual refinement.

### 7.1. Increased Socialization

While pair housing certainly has advantages over single housing, housing NHPs in more complex social arrangements, including small groups, would better promote welfare. Accessibility is often cited as a reason for keeping animals on ID studies in caged housing. However, newer caging designs allow cages to be attached to pens that allow animals to live in groups but still be accessible for research procedures. Monkeys can be trained to enter the cages when access to them is needed. New automated facial recognition technology [65] can help detect the behavior of animals as they are living in groups, which could help predict aggression and thus increase the chances that the animals can live together while on study. 

### 7.2. Awake Blood Draws

Monkeys on ID protocols may need weekly, if not daily, blood draws during certain periods of the study to track viral load as well as to monitor health. NHPs are typically sedated for these blood collections. However, frequent sedations can negatively impact the health of the animals [35,66]. Using PRT to train animals on ID protocols for blood collection is one way to reduce this stress. Macaques can be trained for awake blood draws [46], but this is often not performed on studies in which the pathogen is transmissible through blood, such as SIV, for human safety reasons. Researchers have shown that, while there is an increased human safety risk with awake compared to sedated blood collections, the risk is still quite low; less than 0.2% of awake blood collection events resulted in human exposure [67]. However, this should be assessed by each individual facility. 

### 7.3. Long-Lasting ART

While many of the refinements mentioned have been on the side of clinical and husbandry, advancing the kind of antiretroviral treatment utilized would also be a significant refinement. Currently, the most commonly used treatment for humans with HIV infection is a daily ART pill [68]. However, oral administration is not commonly used in macaque studies, due to the bitter taste of the drug, which is not easily masked in food [69], making the confirmation of medication consumption challenging. Thus, many studies rely on daily subcutaneous injections of ART, which can last for several months. New formulations of antiretrovirals for humans allow for less frequent ART administration [70]. To date, this has not been expanded to NHPs; however, initial pilot studies are underway to assess the feasibility of this option, which would significantly improve animal welfare and staff efficiency for NHP SIV studies.

## Figures and Tables

**Figure 1 vetsci-11-00460-f001:**
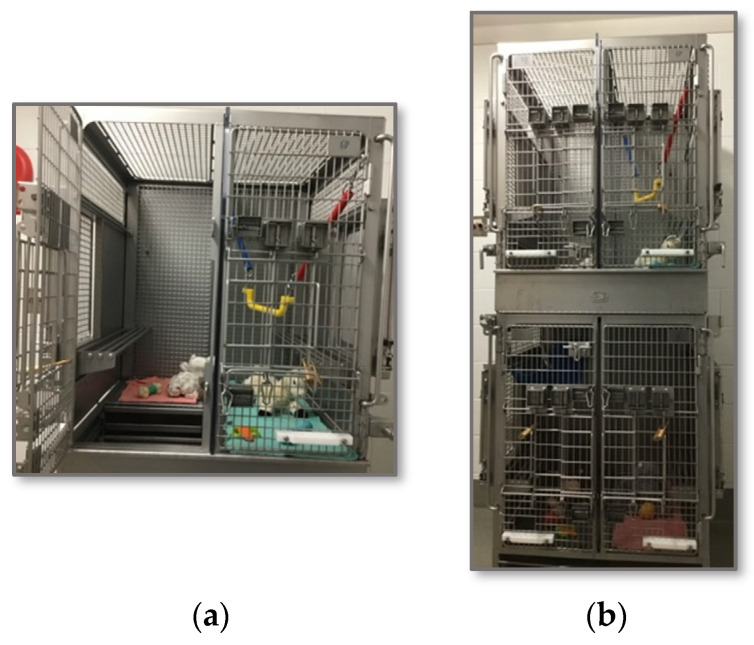
Infant housing for ID studies. Each cage (**a**) contains thermoneutral perching and provides horizontal and vertical access to other cages (**b**), allowing infants to be housed in groups of 4–6.

**Figure 2 vetsci-11-00460-f002:**
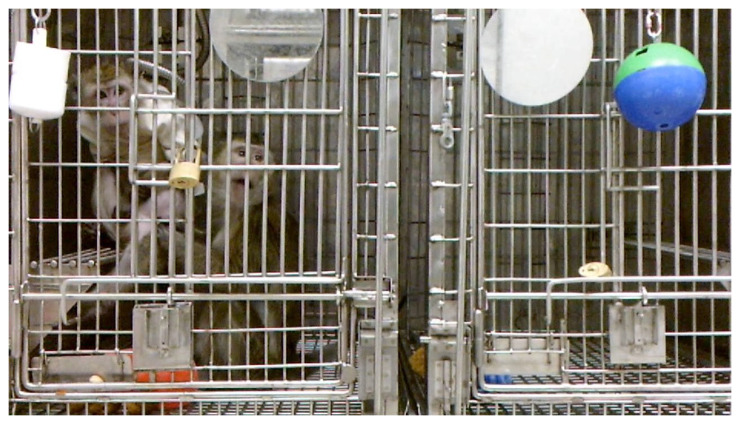
Female cynomolgus macaques pair-housed while one was outfitted with a catheter protection system.

**Figure 3 vetsci-11-00460-f003:**
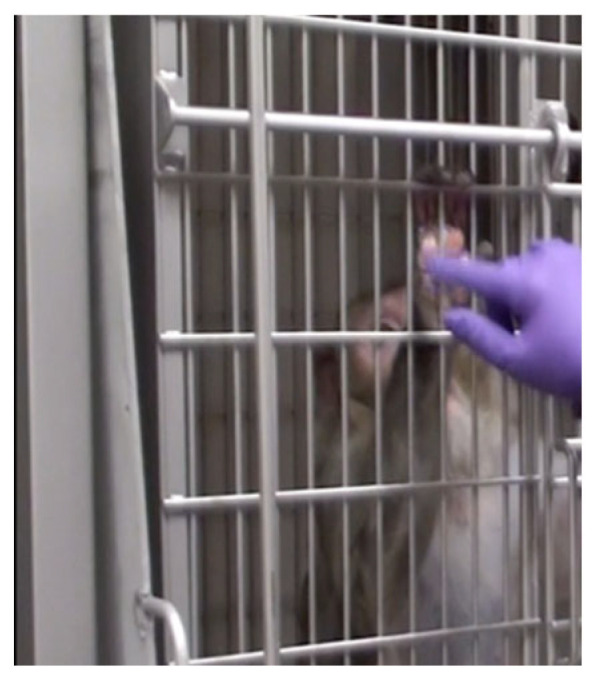
Rhesus macaque trained to allow application of topical ointment.

**Figure 4 vetsci-11-00460-f004:**
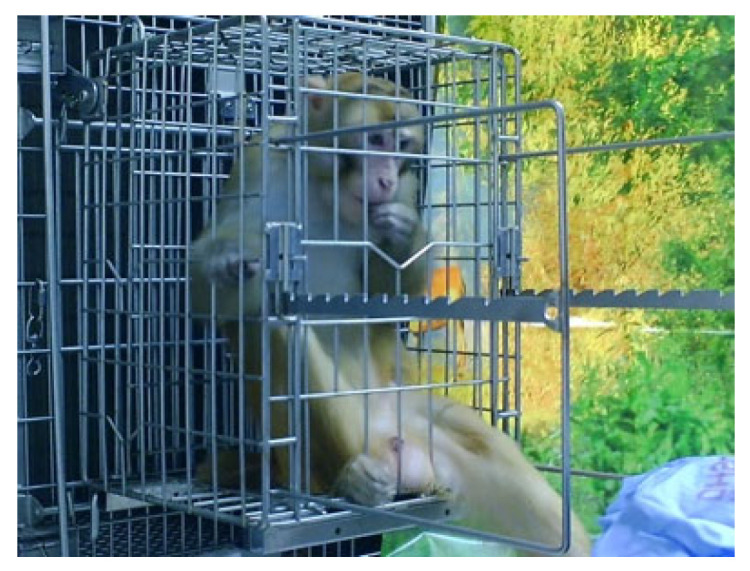
A rhesus macaque in a removable procedure cage. The procedure cage attaches to the exterior of the NHP’s home enclosure. There are holes in the bottom through which the NHP can extend its legs.

**Figure 5 vetsci-11-00460-f005:**
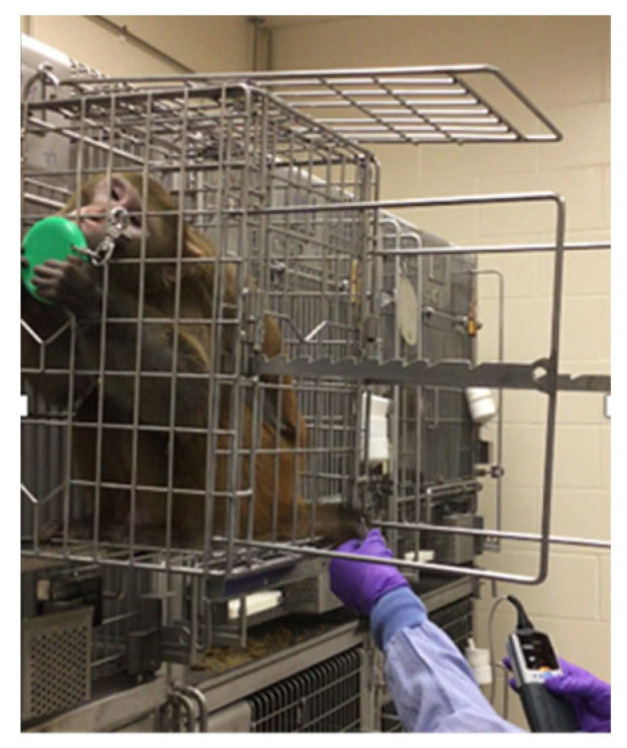
A rhesus macaque trained to enter the procedure cage and allow a pulse oximeter meter on his tail.

## Data Availability

All data are contained within the article.

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
