# Peer review of "Refinements in Clinical and Behavioral Management for Macaques on Infectious Disease Protocols"

_vetsci, 2024, doi:10.3390/vetsci11100460_

Round 1

Reviewer 1 Report

Comments and Suggestions for Authors

This article is a comprehensive, well written review of how innovative behavior management can not only assist veterinary staff in the care of animals but improve animal welfare with direct positive impacts on improving infectious disease science.  The article has many specific points of how more complex living conditions can directly results in better science including less variability, improved outcomes for the animals, easier treatment, decrease staff time for sampling, improved morale for the care takers, etc.  It is easy to read and makes a well thought out argument for socialization in infectious disease research.  The paper specifically addressed common scenarios in infectious disease research such as challenges with SIV, nursery reared infants, catharized and instrumented animals, and frequently sampled animals.  The paper gives very nice, gold standard solutions including housing options for infants, PRT, and step by step ways to work with instrumented animals so that they can be a pair.  I appreciate and would like to highlight the section on infant wearing strategies, implementation of positive reinforcement training, and how even some socialization can result in a significant improvement of the animal.   While this article does call out some challenges with these techniques, in some cases I don’t think they are discussed enough.

There are two areas that I would like to see the authors highlight more within the article.

1.        Employee health challenges associated with some of the proposed techniques and any proposed solutions.  While human safety is addressed with cage side blood draws on animals with SIV I think the human safety considerations should be addressed more thoroughly throughout the paper.  In particular, I think this should be addressed within the positive reinforcement section.  I also think it would be appropriate to discuss how PRT and some of these other techniques minimize future risk to staff.   From my experience, the human safety concerns have a significant impact on how the proposed techniques are viewed and implemented.

2.       The amount of pre-planning needed for these techniques.  While the paper does discuss getting by in and pre-project planning, I think it would be helpful to include generally how long some of these techniques require for full implementation or to highlight implementation prior to study start.  Generally, I think individuals hear about ways to make improvements and want to implement them at the last minute or after an animal has been infected, which may not be the best idea.

Author Response

Reviewer 1

We thank the reviewer for the positive comments about our manuscript.

There are two areas I would like to see the authors highlight more within the article.

  1. Employee health challenges associated with some of the proposed techniques and any proposed solutions. While human safety is addressed with cage side blood draws on animals with SIV I think the human safety considerations should be addressed more thoroughly throughout the In particular, I think this should be addressed within the positive reinforcement section. I also think it would be appropriate to discuss how PRT and some of these other techniques minimize future risk to staff. From my experience, the human safety concerns have a significant impact on how the proposed techniques are viewed and implemented.

 This is an excellent point. We included the increased safety in the PRT section.

  1. The amount of pre-planning needed for these While the paper does discuss getting by in and pre-project planning, I think it would be helpful to include generally how long some of these techniques require for full implementation or to highlight implementation prior to study start. Generally, I think individuals hear about ways to make improvements and want to implement them at the last minute or after an animal has been infected, which may not be the best idea.

Thank you for the suggestion. We have added some details to this effect at the end of section 5.

Reviewer 2 Report

Comments and Suggestions for Authors

Please see attached and below.

This is a timely manuscript that reviews and encourages the use of behavioral management refinements for NHPs on infectious disease studies. As the paper states, it is a thorough, but not necessarily exhaustive, description of strategies that can be used to improve well-being, and reviews the positive impact of such enhancements on research outcomes. It applies the basic concepts of behavioral management (socialization, PRT, behavioral assessment) to a group of NHPs that are in need of such refinements.

In general, I suggest only minor revisions, but these minor things are numerous in quantity. First, the paper would benefit from increased conciseness/less repetition. Second, there is a quite a bit of use of informal language that can be improved by simply using a different word or phrase. Lastly, there are several areas that seem vague and would benefit from additional clarification/definitions. Again, minor things, but should be edited throughout the paper to increase clarity and readability. Below are specific comments regarding the three areas listed above.

Specific comments:

Line 31 (and throughout): Please be consistent in your use of “NHP” vs “NHPs” here and throughout the manuscript. When plural, please use NHPs (e.g., Lines 54, 203, 225, 260, 353, and more).

Line 34: Is there a word missing between “behaviorally depending”?

Line 37 and throughout: Use of commas. Please insert comma after “others” and before “as infection...” to increase readability. There are many other areas where the additional of commas would increase readability, e.g., line 54-55: around the phrase “thus enhancing overall health”; before all uses of “...such as” and “...including” and “...while”; line 254: before “thus”; etc.)

Line 40: is there a citation that can provided for this?

Line 42 and throughout: Please use “singly” rather than “single” when referring to “singly housed” animals. Singly housed is more consistent with socially housed.

Line 63: suggest replacing “yet” with “although”,

Line 69: add “(Macaca sp.)” after “research macaques”.

Line 72: Can you please define “enhanced”? Does this mean increased responsiveness? Is that a good thing?

Line 83: Instead of using the word “fight,” it may be better to say something more research-oriented/descriptive, such as “during agonistic encounters” or “physical altercations.” But it is also important to note that fighting isn’t the only activity that risks transmission, as transmission through saliva can occur through grooming and other positive interactions, as well as through a shared lixit/water source and shared enrichment.

Line 83-86: this sentence is quite long, suggest rewording.

Line 95: delete comma at end of line and put species name in parentheses.

Line 113-120: I think this paragraph would fit better if it followed the paragraph below it. The paragraph that currently starts on line 121 (“Even relatively short periods of...”) continues the line of research regarding benefits of pair housing that is outlined in the paragraph ending on line 112.

Additionally, I think the topic sentence of the paragraph starting on line 113 (that I’m suggesting gets moved down) is a bit off-topic. Currently, the topic sentence doesn’t transition from what has been reviewed in the previous paragraphs, and instead, refers to mitigation of effects of experimental and environmental stressors. I might suggest rewording to “Not only does social housing seems to affect translational value and reproducibility, it also helps reduce...” I think this will increase flow/connection between paragraphs.

Line 118: can you provide a citation for this, as well as in line 120?

Line 125-126: the three uses of “were” should be “was” since the subject is a group (singular).

Line 131: Recommend changing “...provide a buffer to single-housed animals” to “...provide a buffer to the stress experienced by singly-housed animals.”

Line 133: Again, regarding the use of the word “fighting,” perhaps reword to say “Incompatibility between cage mates can lead to agonism and wounding, ...”

Line 138: It seems that identifying the animals wouldn’t be the biggest challenge regarding time, but rather the need to separate the animal from its cage-mate and then administer the treatment? If that is the case, suggest re-working these last two sentences to include that info.

Line 161: “end-up” feels informal. Perhaps revise to “reduces the potential for single housing...”

Line 160-162: I’m not sure I understand what is being communicated here. Please expand. How exactly does housing infants in small groups reduce this potential?

Line 179: figure caption says “or more”. Please be more specific.

Figures 1-5: can pictures be made larger?

Line 188: Can a less vague word than “untoward” be used here to be more descriptive?

Line 188: suggest moving this sentence to the next paragraph where catheterization is the topic. Either worked into the beginning, or perhaps before the sentence starting on line 197. Additionally, are there “few reports,” or are there no reports? If there are a few reports, please cite them. If the only report is the one described in the following paragraphs, please state that.

Line 191: please define “CAR T”

Line 193: suggest changing “as it did” to “as was the case”

Line 245: suggest putting sentence into parentheses following the preceding sentence, and then moving the following paragraph up to read as follows: “...when the specific behavior has been completed (see Pryor [40] and Laule, Bloomsmith [41] for reviews). The use of PRT is recognized as an important...” so that it is one continuous paragraph.

Line 258: please define or describe the ways that it is “easier” to work with trained animals, and in what ways this helps staff. Does this refer to routine husbandry, clinical care?

Line 272: these last two sentences feel like more info than is needed. I would suggest condensing the information. For example, starting on line 272: “...has shown that PRT can mitigate stress associated with SIV infection in pigtailed and rhesus macaques, including lower cortisol, reduced viral loads....” Please also note that the species names for these two were already defined previously, so they do not need to be added here again.

Line 285: Can a citation be added to support the presence of these conditions in ID studies?

Line 288: “body part” should be plural (i.e., “parts”)

Line 318: Can you please be more specific about the type of restraint typically used for adults and why it doesn’t work well for small infants? Are you referring to cage squeezes? Is it a safety issue for the staff or the animals (or both)?

Line 325: Remove “training” after “PRT” (the word “training” is included in the acronym)

Line 342: “setting up” is a bit informal. What about “...assigned to the right project, thus increasing the chances of success for each individual.” ?

Line 352-355: This feels redundant/repetitive with the information presented in the paragraph above. Suggest editing for conciseness.

Line 365: Suggest changing “how” to “ways”.

Line 367 and 400: “Boots on the ground” is another example of informal language that, in my opinion, should be changed to a more appropriately descriptive term.  What about “on-site staff,” “operational staff,” “animal-facing staff.” It may also be helpful to more clearly define these staff at the outset: do you mean animal technicians specifically? Or does this refer collectively to ALL staff that includes those mentioned in lines 402-403? Please be more specific in Line 367 to clarify this.

Line 387: “as needed” rather than “a needed”.

Line 391: informal language: suggest changing “reaching the same end-goal” to “achieving study goals.”

Line 408: Suggest adding “For example,” in front of “Facilities staff may have...” to increase readability.

Paragraphs in line 370-409: Can these paragraphs be tied more closely to ID research/animals? These are all great points, but feel very general in that they apply to all research, not just ID research. It may be helpful to clearly delineate how these strategies can be implemented specifically in ID studies and animals. This could be accomplished by simply adding a few examples specific to ID studies. As just one example, line 389-390: a specific example about how certain strategies (which strategies?) may increase exposure to staff (in what way?), and how EHS may assist with that?

Line 438: Delete “for animals”

Line 453-455: For conciseness and readability, consider revising these lines to read as follows:  “Researchers have shown that, while there is an increased human safety risk with awake compared to sedated blood collections, the risk is still quite low, as less than 0.2% of awake blood collection events resulted in human exposure. However, this should be assessed...”

Lines 401 and 470: “buy-in” is another example of informal language. Consider revising to “support,” “stakeholder engagement,” “stakeholder commitment” etc.

Line 472: The sentence starting on this line and ending on line 475 is awkward and difficult to follow. It feels like there are several ideas that don’t necessarily flow here. Please revise.

Line 477: use “PRT” instead of spelling out.

Line 481: change “where” to “in which”.

Comments on the Quality of English Language

The paper would benefit from minor edits to English, including increased use of commas, revision of informal language, and revision to increase conciseness. 

Author Response

Reviewer 2

We thank the reviewer for their thorough read of our MS and suggestions for improvement.

Line 31 (and throughout): Please be consistent in your use of “NHP” vs “NHPs” here and throughout the manuscript. When plural, please use NHPs (e.g., Lines 54, 203, 225, 260, 353, and more).

We changed NHP to NHPs where appropriate.

Line 34: Is there a word missing between “behaviorally depending”?

Thank you for catching that mistake. We have added “challenging”.

Line 37 and throughout: Use of commas. Please insert comma after “others” and before “as infection...” to increase readability. There are many other areas where the additional of commas would increase readability, e.g., line 54-55: around the phrase “thus enhancing overall health”; before all uses of “...such as” and “...including” and “...while”; line 254: before “thus”; etc.)

We added commas, and also realized we used ‘such as’ a bit too much, so we changed some instances of that phrase.

Line 40: is there a citation that can provided for this?

This is something we have seen from years caring for these animals.

Line 42 and throughout: Please use “singly” rather than “single” when referring to “singly housed” animals. Singly housed is more consistent with socially housed.

We have changed “single housed” to “singly housed” where appropriate.

Line 63: suggest replacing “yet” with “although”,

We have made this suggestion.

Line 69: add “(Macaca sp.)” after “research macaques”.

We added the suggested text.

Line 72: Can you please define “enhanced”? Does this mean increased responsiveness? Is that a good thing?

We combined this sentence with the previous one and made it clear that the immune function is different between single and paired NHPs.

Line 83: Instead of using the word “fight,” it may be better to say something more research-oriented/descriptive, such as “during agonistic encounters” or “physical altercations.” But it is also important to note that fighting isn’t the only activity that risks transmission, as transmission through saliva can occur through grooming and other

positive interactions, as well as through a shared lixit/water source and shared enrichment.

We changed fight to agonistic encounter. We were only providing an example, so we didn’t add other options, although you bring up a good point.

Line 83-86: this sentence is quite long, suggest rewording.

We reworded and shortened this sentence.

Line 95: delete comma at end of line and put species name in parentheses.

We made this suggestion.

Line 113-120: I think this paragraph would fit better if it followed the paragraph below it. The paragraph that currently starts on line 121 (“Even relatively short periods of...”) continues the line of research regarding benefits of pair housing that is outlined in the paragraph ending on line 112.

Additionally, I think the topic sentence of the paragraph starting on line 113 (that I’m suggesting gets moved down) is a bit off-topic. Currently, the topic sentence doesn’t transition from what has been reviewed in the previous paragraphs, and instead, refers to mitigation of effects of experimental and environmental stressors. I might suggest rewording to “Not only does social housing seems to affect translational value and reproducibility, it also helps reduce...” I think this will increase flow/connection between paragraphs.

Thank you for the recommendation. We made this change.

Line 118: can you provide a citation for this, as well as in line 120?

As we changed this paragraph, we deleted one of these lines. We added a reference to the second.

Line 125-126: the three uses of “were” should be “was” since the subject is a group (singular).

Thanks for catching this; we made this suggestion.

Line 131: Recommend changing “...provide a buffer to single-housed animals” to “...provide a buffer to the stress experienced by singly-housed animals.”

We made this suggestion.

Line 133: Again, regarding the use of the word “fighting,” perhaps reword to say “Incompatibility between cage mates can lead to agonism and wounding, ...”

We have changed to “Cage-mates can cause injury …”

Line 138: It seems that identifying the animals wouldn’t be the biggest challenge regarding time, but rather the need to separate the animal from its cage-mate and then administer the treatment? If that is the case, suggest re-working these last two sentences to include that info.

We don’t typically separate the partners for treatment at our facilities, but we did add a line in to that effect. PRT is still a way to reduce that need.

Line 161: “end-up” feels informal. Perhaps revise to “reduces the potential for single housing...”

We made this suggestion.

Line 160-162: I’m not sure I understand what is being communicated here. Please expand. How exactly does housing infants in small groups reduce this potential?

We tried to make this sentence more clear by changing it to “Importantly, transmission of SIV has not been observed in infants housed with 4-5 similarly aged conspecifics (N Haigwood, personal communication) suggesting that group housing of young macaques on ID protocols is possible.” If infants are paired, and one gets sick, the partner may need to be singly housed for the remainder of the study. Housing animals in small groups as opposed to pairs reduces the chances that one animal might become single housed for that reason (i.e., if one animal gets sick and needs to be removed from the study, there are still 4-5 other animals).

Line 179: figure caption says “or more”. Please be more specific.

We removed “or more”. The cages do allow for larger groups, but we don’t recommend that.

Figures 1-5: can pictures be made larger?

We have made the pictures larger.

Line 188: Can a less vague word than “untoward” be used here to be more descriptive?

We changed to “adverse”.

Line 188: suggest moving this sentence to the next paragraph where catheterization is the topic. Either worked into the beginning, or perhaps before the sentence starting on line

  1. Additionally, are there “few reports,” or are there no reports? If there are a few

reports, please cite them. If the only report is the one described in the following paragraphs, please state that.

We modified that sentence, but feel that the paragraph needs a sentence at the end that discusses indwelling catheters, to help segue into the next paragraph.

Line 191: please define “CAR T”

We defined CAR T

Line 193: suggest changing “as it did” to “as was the case”

We made this suggestion.

Line 245: suggest putting sentence into parentheses following the preceding sentence, and then moving the following paragraph up to read as follows: “...when the specific behavior has been completed (see Pryor [40] and Laule, Bloomsmith [41] for reviews). The use of PRT is recognized as an important...” so that it is one continuous paragraph.

We made this suggestion.

Line 258: please define or describe the ways that it is “easier” to work with trained animals, and in what ways this helps staff. Does this refer to routine husbandry, clinical care?

We changed this sentence a bit.

Line 272: these last two sentences feel like more info than is needed. I would suggest condensing the information. For example, starting on line 272: “...has shown that PRT can mitigate stress associated with SIV infection in pigtailed and rhesus macaques, including lower cortisol, reduced viral loads.    ” Please also note that the species names for these

two were already defined previously, so they do not need to be added here again.

We made these suggestions.

Line 285: Can a citation be added to support the presence of these conditions in ID studies?

This finding is based on our years of experience caring for animals on ID protocols.

Line 288: “body part” should be plural (i.e., “parts”)

We changed body part to body parts.

Line 318: Can you please be more specific about the type of restraint typically used for adults and why it doesn’t work well for small infants? Are you referring to cage squeezes? Is it a safety issue for the staff or the animals (or both)?

We added some text to explain that the restraint we were referring to was the cage squeeze. The safety issue is mostly for humans.

Line 325: Remove “training” after “PRT” (the word “training” is included in the acronym)

We removed ‘training’.

Line 342: “setting up” is a bit informal. What about “. assigned to the right project, thus

increasing the chances of success for each individual.” ?

Setting animals up for success is a term commonly used by animal trainers. We would prefer to keep this terminology.

Line 352-355: This feels redundant/repetitive with the information presented in the paragraph above. Suggest editing for conciseness.

We agree, and removed some of the sentences and combined this paragraph with the preceding one to reduce redundancy.

Line 365: Suggest changing “how” to “ways”.

We changed “how” to “ways”.

Line 367 and 400: “Boots on the ground” is another example of informal language that, in my opinion, should be changed to a more appropriately descriptive term. What about “on- site staff,” “operational staff,” “animal-facing staff.” It may also be helpful to more clearly define these staff at the outset: do you mean animal technicians specifically? Or does this refer collectively to ALL staff that includes those mentioned in lines 402-403? Please be more specific in Line 367 to clarify this.

We have changed “boots on the ground” to “animal care staff” the first time it is mentioned, and to “staff” the second time (since there is a list of personnel in the following sentence).

Line 387: “as needed” rather than “a needed”.

We changed ‘a’ to ‘as’.

Line 391: informal language: suggest changing “reaching the same end-goal” to “achieving study goals.”

We made this suggestion.

Line 408: Suggest adding “For example,” in front of “Facilities staff may have...” to increase readability.

We made this suggestion.

Paragraphs in line 370-409: Can these paragraphs be tied more closely to ID research/animals? These are all great points, but feel very general in that they apply to all research, not just ID research. It may be helpful to clearly delineate how these strategies can be implemented specifically in ID studies and animals. This could be accomplished by simply adding a few examples specific to ID studies. As just one example, line 389-390: a specific example about how certain strategies (which strategies?) may increase exposure to staff (in what way?), and how EHS may assist with that?

We included additional examples.

Line 438: Delete “for animals”

We made this suggestion.

Line 453-455: For conciseness and readability, consider revising these lines to read as follows: “Researchers have shown that, while there is an increased human safety risk with awake compared to sedated blood collections, the risk is still quite low, as less than 0.2% of awake blood collection events resulted in human exposure. However, this should be assessed...”

Thank you for the suggestion, which we have made.

Lines 401 and 470: “buy-in” is another example of informal language. Consider revising to “support,” “stakeholder engagement,” “stakeholder commitment” etc.

We made this suggestion.

Line 472: The sentence starting on this line and ending on line 475 is awkward and difficult to follow. It feels like there are several ideas that don’t necessarily flow here. Please revise.

We revised this section.

Line 477: use “PRT” instead of spelling out.

We made this suggestion.

Line 481: change “where” to “in which”.

We made this suggestion.